# Retention strategies among those on community supervision in the South: Lessons learned during the COVID-19 pandemic

**Breana J. Uhrig Castonguay**[1,2]*, **Katherine LeMasters**[2,3], **Chris Corsi**[2], **Evan J. Batty**[4], **Taylor J. Krajewski**[5,6], **Madelene Travis**[2], **Craig Waleed**[7], **Carrie B. Oser**[8], **Kathryn M. Nowotny**[9], **Lauren Brinkley-Rubinstein**[2]

1 School of Medicine, University of North Carolina at Chapel Hill, Chapel Hill, North Carolina, United States of America, 2 School of Social Medicine, Center for Health Equity Research, University of North Carolina at Chapel Hill, Chapel Hill, North Carolina, United States of America, 3 Department of Epidemiology, University of North Carolina at Chapel Hill, Chapel Hill, North Carolina, United States of America, 4 Department of Sociology, University of Kentucky, Lexington, Kentucky, United States of America, 5 Department of Biostatistics, Gillings School of Global Public Health, University of North Carolina at Chapel Hill, Chapel Hill, North Carolina, United States of America, 6 Center for AIDS Research, School of Medicine, University of North Carolina at Chapel Hill, Chapel Hill, North Carolina, United States of America, 7 Disability Rights of North Carolina, Raleigh, North Carolina, United States of America, 8 Department of Sociology, Center on Drug & Alcohol Research, Center for Health Equity Transformation, University of Kentucky, Lexington, Kentucky, United States of America, 9 Department of Sociology and Criminology, University of Miami, Miami, Florida, United States of America

* BJUhrig@gmail.com

## Abstract

### Objectives

Cohort studies must implement effective retention strategies to produce internally valid and generalizable results. Ensuring all study participants are retained, particularly those involved in the criminal legal system, ensures study findings and future interventions will be relevant to this group, who are often lost to follow-up: critical to achieving health equity. Our objective was to characterize retention strategies and describe overall retention among an 18-month longitudinal cohort study of persons on community supervision prior to and during the COVID-19 pandemic.

### Methods

We implemented various retention strategy best-practices (e.g., multiple forms of locator information, training study staff on rapport building, study-branded items). During the COVID-19 pandemic, we developed and describe new retention strategies. We calculated overall retention and analyzed differences between those retained and lost to follow-up by demographic characteristics.

### Results

Prior to the start of the COVID-19 pandemic, 227 participants enrolled across three sites (N = 46 North Carolina; N = 99 Kentucky; N = 82 Florida). Of these, 180 completed the final 18-

**Data Availability Statement:** All relevant data are within the manuscript and its Supporting information files.

**Funding:** This work is supported by the National Institute on Minority Health and Health Disparities (NIMHD) [R01MD013573], The University of North Carolina at Chapel Hill Center for AIDS Research (CFAR), a National Institutes of Health (NIH) funded program P30 AI050410, National Institute of Environmental Health Sciences (NIEHS) (T32ES007018) and The Carolina Population Center provided general and training support (P2C-HD050924; T32-HD007168) from the Eunice Kennedy Shriver National Institute of Child Health and Human Development (NICHD). This work was also supported by NIMHD (F31MD017136) and National Institutes on Drug Abuse (NIDA) – Lifespan/Brown Criminal Justice and Research Training Program (R25DA037190). The funders had no role in study design, data collection and analysis, decision to publish, or preparation of the manuscript.

**Competing interests:** The authors have declared that no competing interests exist.

month visit, 15 were lost to follow-up, and 32 were ineligible. This resulted in an overall retention of 92.3% (180/195). While most participant characteristics did not differ by retention status, a greater proportion of those experiencing unstable housing were lost to follow-up.

## Conclusion

Our findings highlight that when retention strategies are flexible, particularly during a pandemic, high retention is still achievable. In addition to retention best-practices (e.g., frequent requests for updated locator information) we suggest other studies consider retention strategies beyond the study participant (e.g., paying participant contacts) and incentivize on-time study visit completion (e.g., providing a bonus when completed the study visit on time).

## Introduction

Over six million individuals are under criminal legal (CL) supervision in the United States (US) [1]. Over two-thirds of these individuals are not incarcerated in a prison or jail, but rather under a form of community supervision, such as probation or parole. Black, Latinx, and American Indian individuals are substantially over-represented in the CL system due to the system's roots in structural racism [2]. Vulnerable populations, such as those experiencing poverty, mental illness, and substance use disorders, are disproportionately represented and often encounter the CL system at multiple points in their lives [3]. Simultaneously, those with CL involvement, and their families and communities, have disproportionately poor physical and mental health [4–6].

To understand and ultimately mitigate the relationship between the criminal legal system and health inequity, it is critical to retain those with complex criminal legal involvement in research studies. Losing these individuals to follow-up results in selection bias, exacerbates existing inequity in research representation, and prohibits the implementation of programs that can have optimal effects [7]. However, retaining CL-involved individuals in research studies has proven challenging due to competing priorities (e.g., finding employment), poverty, unstable housing, inconsistent contact information, and substance use [7–9].

### What do we know about retention among vulnerable populations?

Historically, research was designed with little to no input from racial and ethnic minorities and other groups experiencing health inequities. Mistrust around research participation is rooted in structural racism and justified hesitancy to participating and staying engaged in research studies [10]. In a meta-analysis of 165 studies focused on recruitment and/or retention techniques of low-income or minority populations, Nicholson and colleagues (2015) found only 15 (9%) studies focused on retention [11]. Of these, language barriers and mistrust were cited as common barriers to retention. Study retention is important for ensuring both internal validity [12] and generalizability of results [13]. For example, a study among substance-using adolescents found those difficult to retain were significantly more likely to report poorer outcomes at their final visit compared to study participants that were more easily retained [14].

Prior work has found the following to be critical for retaining populations facing systemic barriers in research studies, including those with criminal legal involvement: 1) collecting and

tracking detailed locator information, 2) developing strong rapport between research staff and participants, 3) minimizing barriers to participation (e.g. providing study incentives), and 4) being persistent and flexible in retention efforts [15].

Locator information is most helpful when it includes contact and social media information for the participant and close friends, family, or health/social service providers [7, 9, 15, 16]. In addition, identifying physical locations in the community that participants frequent (e.g., community organizations or shelters) is important when locator information changes frequently [7], which is common among individuals involved in the CL system [7–9].

It is also important to build rapport between study staff and participants. This includes building a trusting environment by pairing participants with the same staff member throughout the study [15]. Studies also need to build flexible environments in which staff members are familiar with participants' barriers to continued participation and can accommodate their needs [15, 17]. As the time and resources required to participate in on-going study activities may act as a barrier to retention, staff being flexible on the timing of study visits (e.g., evenings, weekends), offering virtual options, and working with participants to address their specific needs (e.g., childcare) is crucial [8, 9, 15]. Finally, researchers have found success in retention by increasing the persistence and variety of communications with participants [7, 17].

## COVID-19 has created additional challenges to retention

Retention strategies have been complicated by the COVID-19 pandemic, a pandemic that has also disproportionately affected those with systemic barriers to research participation. In this paper, we characterize the implementation of retention strategies among those on community supervision prior to and during the COVID-19 pandemic. To assist with future observational cohorts, we report on lessons learned, share our retention scripts and mailings, and provide training materials to support retention activities. Further, beyond research studies, these retention activities can support the clinical care continuum for illnesses that disproportionately affect people involved in the CL system (e.g., achieving viral suppression for HIV, substance use disorder, Hepatitis C) [18–23].

To our knowledge, this is the first report detailing strategies to enhance retention among populations during the COVID-19 pandemic era.

## Materials and methods

### Study design

The Southern Pre-Exposure Prophylaxis (PrEP) Study (SPECS) is an 18-month prospective longitudinal cohort of individuals with CL involvement and a clinical indication for PrEP in North Carolina (NC), Florida (FL), and Kentucky (KY). Its goal is to understand how individual, social, and structural factors exacerbated by CL system involvement affect PrEP knowledge, acceptability, initiation, and sustained use. It has been described in detail elsewhere [24]. Study participants provided written consent and this study was approved by the institutional review boards at the University of North Carolina at Chapel Hill (18–2466). Recruitment began on June 13, 2019, and was paused on March 15, 2020 due to the COVID-19 pandemic. Recruitment re-started on February 26, 2021, with the three sites returning to in-person recruitment and retention at variable dates. This analysis focuses on individuals recruited prior to the COVID-19 pandemic and retention efforts with this first wave of our cohort. Study assessments occurred at baseline, 6-, 12-, and 18-months, and phone check-ins with participants occurred between study visits (Fig 1). Phone check-ins were for retention purposes only.

## Pre-Covid Retention Strategies

**Fig 1. Study assessment schedule and retention strategies.**

### Retention definitions

The primary outcome of interest for this analysis was study retention at 18 months, the end of follow-up. However, retention was also calculated at 6- and 12-month visits. At each time point (6, 12, 18 months) a missed study visit was defined as any study visit that was not completed within the follow-up visit window. Prior to COVID-19, the follow-up visit window was +/- 2 months. On August 1st, 2020, the follow-up visit window was increased to +/- 3 months to account for the challenges of the COVID-19 shutdown. Study retention was defined as the proportion of study participants who completed the scheduled study visits (in-person or virtual) within the allowable follow-up visit window. Note, participants were allowed to complete the 18-month visit, within the allowable follow-up visit window, regardless of how many other study visits they completed between baseline and 18-months. We did not include in the final 18-month retention status those who were re-incarcerated during the 18-month study visit window or revoked consent or deceased at any point during the study. Thus, loss to follow-up (LTFU) at the 18-month study visit, the primary timepoint of interest, was defined as the proportion of those eligible for the 18-month study visit that did not complete it.

### Retention strategies

Recognizing the unique challenges of study retention among CL-involved populations, study sites used a combination of techniques to prevent loss to follow-up over the 18-month study period. Retention techniques were identified via a combination of the authors' experience with retention in prior studies and an in-depth literature review on retention strategies [25]. All sites implemented the same core retention strategies and monitored and developed site-specific retention activities tailored to their participants. All sites attended a mandatory Retention Training Retreat, focused on five retention categories informed by the pivotal work by Haley et al.: 1) staff capacity building; 2) interpersonal relationship building; 3) participant tracking systems; and 4) community engagement; and 5) reduction of external barriers to participation [15]. We describe below the retention strategies and adjustments made during COVID-19 for these five retention areas and describe in Tables 1 and 2 these strategies by study staff and study participant, respectively.

**Table 1. Retention strategies and changes during COVID-19 for *STUDY STAFF*.**

| Retention Strategy Category | Retention Strategy | Description | How strategy was different during COVID-19 |
|---|---|---|---|
| Staff Capacity Building | Mandatory Retention Trainings | Reviewed and practiced core retention strategies. | Mandatory virtual training (August and December 2020) introduced new Covid-specific retention strategies. |
| | Retention calls | Monthly retention calls for staff to discuss retention "wins" and strategize on difficult retention cases. | Calls increased to twice a month. |
| Interpersonal Relationship Building | Rapport Building Trainings | Reviewed best practices for rapport building. | Discussed virtual rapport building strategies (e.g., increasing holiday cards). |
| | Hiring study staff reflective of study participants | Common backgrounds between staff and study participants builds rapport and establishes trust. | Reviewed with staff members ways to continue to build rapport and reflect on commonalities over the phone. |
| Participant Tracking Systems | White Pages | Online database to obtain additional contact information on hard-to-reach participants. Be aware some addresses and phone numbers may be out of date. | Added after Covid-19 shutdown. |
| | Online prison/jail databases | State-level prison/jail databases (e.g., VINELINK). | Used throughout entire study. |
| Community Engagement | Community-Based Organizations (CBOs) | Prior to study initiation, introduced study (without disclosing eligibility criteria) to CBOs and discussed best ways to reach the study team, e.g., study flyers. If unsure of key CBOs, reassess locator information as study participants enroll. | Study staff completed an audit of new community organizations and places participants may engage with during the pandemic (e.g., any new food banks in the area). Mailed study-branded materials and participant letters to key CBO leaders. |

Retention strategy categories by: Haley DF, Lucas J, Golin CE, Wang J, Hughes JP, Emel L, et al. Retention Strategies and Factors Associated with Missed Visits Among Low Income Women at Increased Risk of HIV Acquisition in the US (HPTN 064). AIDS Patient Care STDS [Internet]. 2014 Apr 1;28(4):206–17.

**Staff capacity building (Table 1).** At the initial staff retention training, staff learned the importance of study retention, reviewed the study's core retention strategies, identified site-specific retention strategies, and practiced case scenarios of difficult retention cases. All sites participated in a monthly phone call to discuss study specific items, including retention. After the COVID-19 shutdown, we increased staff capacity building by adding virtual bi-monthly retention calls to discuss individual retention cases.

**Interpersonal relationship building (Tables 1 and 2).** At training, emphasis was placed on the importance of rapport-building and developing trusting relationships with participants and key stakeholders in study communities. Sites were encouraged to have the same study staff member interact with a participant and birthday and holiday cards were sent to participants.

During COVID-19, we increased mailings to participants (S1 Appendix) and sent packages of study-branded materials. We identified new study-branded items that would be more useful for participants during the pandemic, such as hand sanitizer and at-home exercise equipment (e.g., resistance bands). We also identified study-branded items that might be useful for participant's contacts—particularly important when the only point of contact with a participant was via this contact, such as fridge magnets with space to write memos and the study logo and phone number (S1 Table).

**Participant tracking systems (Tables 1 and 2).** At every study visit, study staff filled out a *new* locator form. The locator form asked for varied information from participants (e.g., current address, housemates) (S1 File). Sites were also encouraged to personalize locator forms to reflect unique site-specific characteristics. Recognizing the potential resistance of study participants providing contact information, study staff participated in role-plays to practice building rapport.

During COVID-19, study staff pivoted to virtual retention, implementing two new strategies. First, we added locator form check-ins and paid participants $5 for contacting study staff

**Table 2. Retention strategies and changes during COVID-19 for *STUDY PARTICIPANTS*.**

| Retention Strategy Category | Retention Strategy | Description | How strategy was different during COVID-19 | Incentive Amount, if applicable |
|---|---|---|---|---|
| **Interpersonal Relationship Building** | **Project logo** | We developed a logo that distinguished our project and did not reveal study eligibility criteria. | We purchased additional logo-branded items to mail to participants. | N/A |
| | **Mailings to participant** | Mailings such as birthday cards and holiday cards that remind participant of the study and their next visit. | We significantly increased our participant mailings when in-person contact was halted. (Covid-specific mailings approved in September 2020) | N/A |
| | **Study-branded items** | Project sites selected items they felt would be beneficial to their study population. All items included the study logo and phone number. Items were also distributed to participant contacts. | New items were identified to hopefully be more useful during a pandemic (e.g., exercise resistance bands with instructions and hand sanitizer) or more visible (e.g., magnetic memo writing pad with dry erase marker). | N/A |
| **Participant Tracking Systems** | **Locator Form** | Asked varied contact information from participants. | Developed script to explain contact incentive. | N/A |
| | **Retention Journal** | A place for study staff to document each encounter with the participant. | During trainings, study staff reminded of the importance of documenting all details. | N/A |
| | **Phone check-ins** | Between each data collection visit (i.e., baseline, 6 mo), there were retention-specific phone check-ins. | Developed script to introduce payment for locator updates. | $10/check-in |
| | **Payment for locator information updates** | Participants encouraged to call study staff to update locator information. Participants with no locator form changes still encouraged to call and "check-in". | Added December 2020 after Covid-19 shutdown. | $5/update (up to 6 times) |
| | **Social platforms (Facebook, etc.)** | Collected information on all social media platforms. | Gained the ability to use social media during the pandemic (July 2021). Required a re-consent of participants. | N/A |
| | **Texting** | Sending texts (SMS) to participants. Easier, particularly for participants using "texting apps" such as Textmail, which allow individuals to text without a cellular plan. | Gained the ability to text message participants during the pandemic (December 2020). Required a re-consent of participants. | N/A |
| | **Study Phone Number** | Study staff had the same study phone number throughout the duration of the study. | New study-branded materials had study contact information. | NA |
| | **Contact incentive** | An incentive to pay a participant's contact (e.g. an uncle) listed on the locator form, if the contact successfully connected study staff to the participant and the participant completed their visit. | Added December 2020 after Covid-19 shutdown. | $15/contact |
| **Community Engagement** | **Field visits** | Visiting participants and their contacts in the field. Important to bring study-branded materials. | We were limited in being able to go into the field during the COVID-19 shutdown. Came up with new health safety measures when deemed safe to go into the community (e.g., staffing driving separate cars). | N/A |
| **Reduction of External Barriers to Participation** | **Incentives** | Participants paid for each study visit and retention phone call. | Added incentives for on-time visits and updating locator information. | |
| | **Venues easy to access/parking** | Provided free transportation (e.g., rideshare programs such as Lyft or bus tokens) when needed or parking passes for in-person participant visits. | During Covid, visits moved to virtual. | N/A |
| | **Childcare/snacks** | Provided toys and activities in the waiting room to support study participant's with children. Provided snacks and beverages. | During Covid, visits moved to virtual. | N/A |
| | **Bonus for "on-time" study visit completion** | To encourage virtual visits to be completed on-time, added bonus when visit completed within 1 week. | Added December 2020 after Covid-19 shutdown. | $5 for each visit completed within +/- 7 days of study visit |

Retention strategy categories by: Haley DF, Lucas J, Golin CE, Wang J, Hughes JP, Emel L, et al. Retention Strategies and Factors Associated with Missed Visits Among Low Income Women at Increased Risk of HIV Acquisition in the US (HPTN 064). AIDS Patient Care STDS [Internet]. 2014 Apr 1;28(4):206–17.

to update their locator information or confirm that there were no changes [26, 27]. These brief check-ins were in *addition* to the retention-specific phone calls scheduled for 3-, 9-, and 15-months. Second, we added a $15 incentive for a participant's contact (e.g., uncle or aunt) if the contact successfully connected study staff to the participant and the participant completed the study visit (Fig 1). We developed new retention scripts to introduce these strategies (S2 Appendix) and gained the ability to message participants via text message and social media, during COVID-19.

**Community engagement (Tables 1 and 2).** Using prior successful community engagement study design models [15, 25], sites identified community partners and events to attend to build relationships with key stakeholders. Focus was placed on community partners study participants would engage with such as organizations focused on supporting CL-involved persons, food banks, and homeless shelters.

**Reduction of external barriers to participation (Table 2).** Prior to COVID-19, to reduce participant burden, same day enrollment was offered to all study participants. Participants were offered visits outside of normal work hours, study site venues were conveniently and centrally located to participants, and participants were provided travel compensation (e.g., bus fare, parking passes) if needed. During the COVID-19 shutdown, study visits were completed by phone instead of in-person. To encourage these virtual visits to be completed on-time, we added a paid $5 "on-time bonus" for participants who completed a data collection visit within one week of their scheduled visit (Fig 1).

Initially participants were paid $190 during the course of the study: $40 for completing in-person visits and $10 for retention check-ins. After adding the COVID-19 additions such as the locator information check-ins and on-time bonus, the monetary amount paid over the course of the study increased from up to $190 to up to $250 per participant.

## Data analysis

Associations between demographic variables and 18-month retention status (retained vs. LTFU) were assessed using chi-square tests or Fisher's exact tests (when sample size was insufficient) for categorical variables and t-tests for continuous variables with Satterthwaite adjustment when variances were unequal. A 0.05 type I error rate was applied with no adjustment for multiplicity. Complete case analyses were conducted and missing data were excluded. Analyses were conducted in Windows SAS version 9.4 (Cary, NC).

## Results

### Sample characteristics

As of March 2020, when recruitment paused due to the COVID-19 pandemic, SPECS had enrolled 227 participants across three sites (N = 46 NC; N = 99 KY; N = 82 FL). Of these, 180 completed the final 18-month visit, 15 were LTFU, and 32 were ineligible (21 were reincarcerated during the 18-month visit window, four revoked consent, and seven were deceased). This resulted in 195 participants eligible for the 18-month visit and an overall retention of 92.3% (180/195). Of the 180 retained, 80/180 (44.4%) were from Kentucky, 68/180 (37.8%) were from Florida, and 32/180 (17.8%) were from North Carolina. Of the 15 LTFU, 10 (66.7%) were from Kentucky, 4 (26.7%) from North Carolina, and 1 (6.7%) from Florida. Study retention at the 6- and 12-month visits gradually increased from 80% to around or above 90% for most sites (Fig 2).

As previously published, this cohort represents a diverse population in the South [28]. The average age of participants was similar among those retained and those LTFU (retained: 36.8 years; LTFU: 33.2 years) (Table 3). The majority of those retained and those LTFU identified

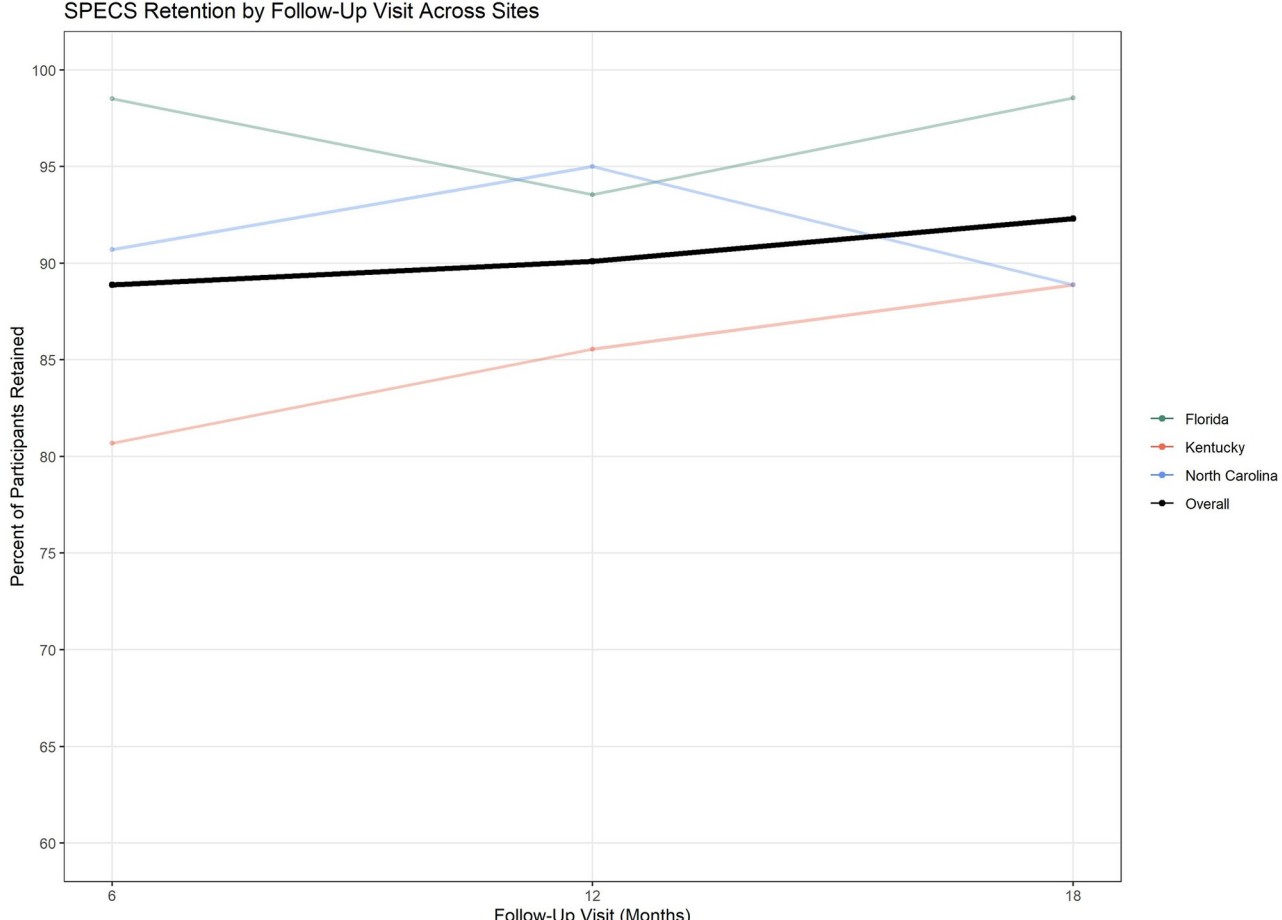

**Fig 2. Retention overtime by site and study visit.**

as men (retained: 67.8%, LTFU: 66.7%). Race/ethnicity did not vary by retention status, nor did most recent length of incarceration. A larger proportion of those LTFU (60.0%) than retained (21.7%) experienced housing instability. A sensitivity analysis combining those LTFU (N = 15) and those ineligible for an 18-month visit (N = 32) showed no significant differences in demographic characteristics compared to those retained (N = 180) (S2 Table).

## Discussion

With an overall retention of 92.3% at the 18-month assessment visit among individuals on community supervision, this cohort study demonstrated high retention in a vulnerable population despite unforeseen barriers due to the COVID-19 pandemic. Further, the above 80% retention across all sites at the 6-month visit highlights the important foundation built by the initial retention strategies. This foundation allowed for the gradual retention increase to near and above 90% once additional strategies were implemented. The lack of differences between retained and LTFU individuals indicates equitable retention and ensures this study's findings and future interventions are relevant to this vulnerable population. However, the higher proportion LTFU experiencing unstable housing speaks to the need to ensure retention strategies are appropriate for these individuals.

**Table 3. Demographics of participants eligible for 18-month visit (N = 195).**

| | 18-month retention status | | | |
| | Complete | Lost to follow up | Total | P Value |
| | (N = 180) | (N = 15) | (N = 195) | |
|---|---|---|---|---|
| **Age** | | | | 0.22 |
| Mean (SD) | 36.8 (11.1) | 33.2 (7.1) | 36.5 (10.9) | * |
| Median (Range) | 35.0 (20.0–71.0) | 32.0 (22.0–45.0) | 35.0 (20.0–71.0) | |
| N (N Missing) | 180 (0) | 15 (0) | 195 (0) | |
| **Gender Identity** | | | | 1.00[α] |
| Man | 122 (67.8%) | 010 (66.7%) | 132 (67.7%) | |
| Woman | 058 (32.2%) | 005 (33.3%) | 063 (32.3%) | |
| Missing | 000 (00.0%) | 000 (00.0%) | 000 (00.0%) | |
| **Race/Ethnicity** | | | | 0.82[α] |
| White Non-Hispanic | 071 (39.4%) | 007 (46.7%) | 078 (40.0%) | |
| Black Non-Hispanic | 067 (37.2%) | 004 (26.7%) | 071 (36.4%) | |
| Hispanic | 032 (17.8%) | 003 (20.0%) | 035 (17.9%) | |
| Other/Unknown | 010 (05.6%) | 001 (06.7%) | 011 (05.6%) | |
| Missing | 000 (00.0%) | 000 (00.0%) | 000 (00.0%) | |
| **Housing Status** | | | | 0.003[α] |
| No permanent, stable place to live | 039 (21.7%) | 009 (60.0%) | 048 (24.6%) | |
| Permanent, stable place to live | 141 (78.3%) | 006 (40.0%) | 147 (75.4%) | |
| Missing | 000 (00.0%) | 000 (00.0%) | 000 (00.0%) | |
| **Time in jail or prison, months (most recent release)** | | | | 0.99 |
| Mean (SD) | 23.3 (52.1) | 23.1 (41.5) | 23.3 (51.3) | |
| Median (Range) | 7.0 (0.0–559.7) | 8.0 (0.2–158.2) | 7.0 (0.0–559.7) | |
| N (N Missing) [β] | 168 (12) | 15 (0) | 183 (12) | |

[α] Fisher's Exact test for p value

[β] 7 participants were not in jail or prison within 12 months prior to their baseline visit

Cohort studies with low retention rates are not internally valid for their study population and limit the potential for future interventions. While study retention is critical, deciding on which retention strategies to deploy before and during the study can be a challenge. Prior work shows that research studies that engage with multiple retention strategies reported higher retention rates than studies that used fewer retention strategies [29]. We used both well-documented retention strategies (e.g., importance of building rapport, locator forms) and less frequently used retention strategies (e.g., contact incentives, on-time bonuses). Furthermore, given our high equitable retention using both well-documented and novel retention strategies, we hope this can be applied to clinical care retention. Prior studies have shown that individuals with CL involvement and other competing needs to achieving good health (e.g., unstable housing) led to poor retention in clinical care—contributing to health inequities [18–21].

There is no formal guidance on what constitutes an acceptable retention rate, particularly within different study designs (e.g., randomized controlled trials, cohorts, etc.) or among varied populations. Some suggest a 5% loss to follow-up (i.e., 95% retention) leads to little bias, but greater than 20% loss poses serious threats to validity [30, 31]. Retention guidance is particularly important when working with vulnerable populations, who are frequently lost to follow-up. Future work is needed to review and synthesize acceptable retention rates within these populations.

## Limitations

While we do believe quality and quantity of retention strategies contributed to retaining 92.3% of our sample, it can be challenging to quantify these efforts, particularly by each retention strategy. Prior work has repeatedly concluded that it is not possible to identify which specific retention strategies were particularly helpful in retention when multiple were used [32]. While calls to action for more evaluative retention data have been made, this has largely not been done. Our retention efforts for this cohort wave (e.g., those that enrolled prior to the COVID-19 pandemic) were conducted primarily via paper and pencil or locked, fillable PDFs. While all retention strategies used for each participant were documented in individual, paper-copy retention journals, the analysis of which retention methods were most successful is too labor intensive to quantify. This is a limitation of this study, as we are unable to analyze the "dose" of retention activities and the amount of time required to retain the most difficult cases. To that end, we have recently developed an app-based recruitment and retention tracking system for our second cohort wave (e.g., those that enrolled in the study after the COVID-19 pandemic began). This platform allows for detailed documentation of contact attempts (i.e., which phone numbers were previously used) and the amount of time spent on each activity by research staff. The platform will allow us to run real-time frequencies of, for example, the number of changes to participant locator information throughout the study and the number of attempts by retention strategy type, (e.g., mailings, phone calls to participant contact) that lead to successful retention. This system will increase data fidelity and contribute to the research gap to quantify retention efforts. Further, the retention field would benefit from qualitative insights from participants as to which strategies most contributed to their study and/or clinical retention.

## Conclusion

Cohort studies require considerable effort and resources, and should be more systematic in understanding [33] where to put those resources to ensure findings are relevant for the entire study population. Herein we report a number of retention strategies that resulted in 92.3% retention in a longitudinal cohort study among a population that typically has low retention, those involved in the criminal legal system. These strategies can be used and built upon by other researchers to ensure study validity and reliable results upon which effective interventions can be built. We hope these retention strategies can also be used to improve clinical care retention among this population. Furthermore, future research that documents the cost effectiveness of retention efforts is urgently needed.

## Supporting information

**S1 Appendix. Study mailers.**
(DOCX)

**S2 Appendix. Retention scripts.**
(DOCX)

**S1 File. Locator form.**
(PDF)

**S1 Table. Study branded retention items.**
(DOCX)

**S2 Table. Demographics of participants who completed the 18-month (N = 180) and who did not complete (N = 47: Ineligible [N = 32] and LTFU [N = 15]).**
(TIF)

## Acknowledgments

This research would not have been possible without the North Carolina Department of Public Safety, Kentucky Department of Corrections, and Florida Department of Corrections participation; however, the findings and ideas presented are solely those of the authors. We would also like to thank Lorin Bruckner for her support with the table visualizations.

## Author Contributions

**Conceptualization:** Breana J. Uhrig Castonguay, Carrie B. Oser, Kathryn M. Nowotny, Lauren Brinkley-Rubinstein.

**Data curation:** Katherine LeMasters, Chris Corsi.

**Formal analysis:** Taylor J. Krajewski.

**Methodology:** Breana J. Uhrig Castonguay.

**Project administration:** Chris Corsi.

**Visualization:** Taylor J. Krajewski.

**Writing – original draft:** Breana J. Uhrig Castonguay, Katherine LeMasters.

**Writing – review & editing:** Katherine LeMasters, Chris Corsi, Evan J. Batty, Taylor J. Krajewski, Madelene Travis, Craig Waleed, Carrie B. Oser, Kathryn M. Nowotny, Lauren Brinkley-Rubinstein.

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
