## [Decision Letter · Decision Letter 0]

16 Jan 2023

PONE-D-22-30143Retention Strategies in the First Pre-Exposure Prophylaxis (PrEP) Observational Cohort among those on Community Supervision in the South: Lessons Learned During the COVID-19 PandemicPLOS ONE

Dear Dr. Uhrig Castonguay,

Thank you for submitting your manuscript to PLOS ONE. After careful consideration, we feel that it has merit but does not fully meet PLOS ONE’s publication criteria as it currently stands. Therefore, we invite you to submit a revised version of the manuscript that addresses the points raised during the review process.

In your revision, please address all of the reviewers' comments.

We look forward to receiving your revised manuscript.

Kind regards,

Douglas S. Krakower, MD

Academic Editor

PLOS ONE

Journal Requirements:

a) Did participants provide their written or verbal informed consent to participate in this study?

4. Please expand the acronym “NIH, NIMHD, NIEHS” (as indicated in your financial disclosure) so that it states the name of your funders in full.

Reviewers' comments:

Reviewer's Responses to Questions

**Comments to the Author**

1. Is the manuscript technically sound, and do the data support the conclusions?

Reviewer #1: Yes

Reviewer #2: Yes

2. Has the statistical analysis been performed appropriately and rigorously? 

Reviewer #1: Yes

Reviewer #2: Yes

3. Have the authors made all data underlying the findings in their manuscript fully available?

Reviewer #1: Yes

Reviewer #2: Yes

4. Is the manuscript presented in an intelligible fashion and written in standard English?

Reviewer #1: Yes

Reviewer #2: Yes

5. Review Comments to the Author

Reviewer #1: This study addresses an important and practical topic relevant to those working with individuals involved in the criminal legal system, both from a research and clinical perspective. Retention in research studies (and clinical care) is critical to the comprehensive understanding of outcomes in this population. The authors provide a well-organized overview of the toolkit that can be used with staff and clients to maximize retention, and include modifications needed during COVID. With regards to the results, they present a remarkably high retention rate with this participant population, especially during a pandemic. It is a missed opportunity to focus only on research studies, as these retention efforts could be critical to the clinical care continuum for many illnesses that are relevant to this patient population—HIV (treatment and prevention), substance use disorder, mental illness, hypertension, etc.—and would recommend broadening the framing of the article as being relevant to both research and clinical care. However, the title as currently written is misleading and would drop the PrEP component as this really is not mentioned in the paper at all. The biggest weakness of this study is that the delivery of specific retention components is not addressed. If the authors have data on this, that should be presented. If every participant received the complete resource-intensive bundle, this should be indicated and recognized as a limitation. What we learn then is that implementing all of these best practices simultaneously can work, but does not help guide those with limited resources as to how best to invest them.

Abstract:

It is odd that the abstract does not mention PrEP at all. It is also not mentioned in the introduction and since PrEP is really not the focus of the paper, I would remove it from the title

Intro

Lines 64-66 Although it is understandable that the authors, as researchers focus on retention in research studies, what about retention in clinical care? What about the relationship between retention in study and retention in care?

Tables 1 and 2

Thorough but quite large, may be beneficial to fit each one onto one page

Methods

Primary outcome is retention at 18 months, did virtual /phone visits count for this outcome?

All participants were on community supervision, was there any interaction with the study team and probation/parole officers?

Results

It would be nice to know which retention methods were deployed and how often. Presumably the staff retention training and methods occurred for all staff, but did all participants receive all the retention strategies? As these are labor-intensive endeavors, it would be helpful to know what was truly required to achieve this level of retention. This could help guide future efforts, whether they be in research or for the purposes of providing healthcare or other post-incarceration services, in order to estimate what level of resources are needed to provide these services. Also, some individuals may need less, some may need more engagement to be retained and it would be useful to understand this spectrum. This is reminiscent of a “bundled” approach to quality improvement where it is difficult to tease out what was most effective.

Did everyone start out with the same level of retention strategies deployed and then adapted based on need or was is consistent throughout regardless of engagement? The penultimate paragraph of the discussion (is this a limitations paragraph?) addresses this to some degree but would be helpful to describe in methods and/or results how these strategies were deployed to the extent that this was measured.

Discussion

Would benefit from having an actual limitations paragraph. If the authors are not able to measure which retention strategies were deployed when or some type of “dose” of retention intervention, then this needs to be explicitly stated as a limitation. Also, it may have been helpful to hear from participants what they felt contributed most to retention (either through a survey or qualitative interviews), as this too could help guide future efforts.

Reviewer #2: This is an important contribution to a growing body of work that highlights the value of including people with criminal-legal (CL) involvement in research, and strategies to keep participants engaged in the midst of a global pandemic. This is particularly important for populations with overlapping disparities, such as those with CL involvement. I strongly recommend publication, however I have a few suggestions for revisions below:

1) Introduction (Line 72): You discuss retention strategies for criminal-legal (CL) involved individuals, however I wonder if you can include some reasons for loss to follow-up as well (for example, historical mistrust of researchers). Additionally, a brief discussion of trends in retention for different sociodemographic/geographic groups (or lack thereof) would be helpful as this is a component of your analysis.

2) Line 147: What informed your decision to use these specific retention strategies? Including the citation from Table 1 would be helpful here that provides some context and reasoning.

3) Line 151: Generally, across what time period were changes to your retention strategies implemented during the COVID-19 pandemic? As written, the reader is left to infer it was around the time of March 2020. Including a timeline provides more support for your claim that these strategies contributed to a high retention rate.

4) Tables 1 and 2: Different verb tenses and writing styles are used throughout the table in an inconsistent manner. I recommend streamlining them to reflect an objective description of the strategies throughout. I also recommend removing any strategies that were not used in this study from the table instead of denoting them as "Strategy not used in this research study". Strategies that were not used can be disclosed in the manuscript text, however I don't think it is necessary.

6. PLOS authors have the option to publish the peer review history of their article (what does this mean?). If published, this will include your full peer review and any attached files.

Reviewer #1: **Yes: **Ank Nijhawan

Reviewer #2: No

---

## [Author Response · Author response to Decision Letter 0]

7 Mar 2023

Updated edits from 3/3/23 email: 

1. Can you please upload an additional copy of your revised manuscript that does not contain any tracked changes or highlighting as your main article file. This will be used in the production process if your manuscript is accepted. Please amend the file type for the file showing your changes to Revised Manuscript w/tracked changes. Please follow this link for more information: http://blogs.PLOS.org/everyone/2011/05/10/how-to-submit-your-revised-manuscript/

Apologies for this issue. New manuscript with no track changes uploaded (Manuscript) and an updated “Revised Manuscript w/tracked changes” uploaded too. 

2. Please expand the acronym “NIH, NIMHD, NIEHS” (as indicated in your financial disclosure) so that it states the name of your funders in full. 

This has been added to the cover letter. I cannot find the “financial disclosure section” to address the other comment. 

3. Please note that funding information should not appear in the Acknowledgments section or other areas of your manuscript. We will only publish funding information present in the Funding Statement section of the online submission form. Please remove any funding-related text from the manuscript. 

Thank you. This has been removed from the manuscript. 

4. Your image file "Fig2.jpeg" cannot be opened and processed, the image file is corrupt or invalid. You will need to convert your image file to another format or fix the current image, then re-submit it. 

This has been converted to a PDF and uploaded. 

Edits as follows: 

Made changes to font size, bold for level 1, 2, and 3 titles based on the formatting guidelines. 

Removed italics from headers 

Added Supporting information at the end 

a) Did participants provide their written or verbal informed consent to participate in this study? 

Participants did provide written consent. We added the following line to the methods section: “Participants provided written consent and this study was approved by the institutional review boards at the University of North Carolina at Chapel Hill (18–2466).” 

The information in both sections (Funding information and Financial Disclosures) now matches. 

4. Please expand the acronym “NIH, NIMHD, NIEHS” (as indicated in your financial disclosure) so that it states the name of your funders in full. This information should be included in your cover letter; we will change the online submission form on your behalf. 

Thank you for making these changes in the cover letter 

We have expanded the acronyms within the funding section to state the name of our funders in full. 

This has been completed, supporting information is now at the end of the manuscript and in-text citations match. 

This is complete. We’ve added the following statement to the Methods Section: “Study participants provided written consent and this study was approved by the institutional review boards at the University of North Carolina at Chapel Hill (18–2466).” 

We’ve reviewed and added references. 

Reviewer 1: 

Reviewer #1: This study addresses an important and practical topic relevant to those working with individuals involved in the criminal legal system, both from a research and clinical perspective. Retention in research studies (and clinical care) is critical to the comprehensive understanding of outcomes in this population. The authors provide a well-organized overview of the toolkit that can be used with staff and clients to maximize retention, and include modifications needed during COVID. With regards to the results, they present a remarkably high retention rate with this participant population, especially during a pandemic. It is a missed opportunity to focus only on research studies, as these retention efforts could be critical to the clinical care continuum for many illnesses that are relevant to this patient population—HIV (treatment and prevention), substance use disorder, mental illness, hypertension, etc.—and would recommend broadening the framing of the article as being relevant to both research and clinical care. However, the title as currently written is misleading and would drop the PrEP component as this really is not mentioned in the paper at all. The biggest weakness of this study is that the delivery of specific retention components is not addressed. If the authors have data on this, that should be presented. If every participant received the complete resource-intensive bundle, this should be indicated and recognized as a limitation. What we learn then is that implementing all of these best practices simultaneously can work, but does not help guide those with limited resources as to how best to invest them. 

Abstract: 

It is odd that the abstract does not mention PrEP at all. It is also not mentioned in the introduction and since PrEP is really not the focus of the paper, I would remove it from the title 

We thank the reviewer for their comments. We’ve removed PrEP from the title and only mention PrEP within the methods section. 

Intro 

Lines 64-66 Although it is understandable that the authors, as researchers focus on retention in research studies, what about retention in clinical care? What about the relationship between retention in study and retention in care? 

The Reviewer makes a very interesting point, and we thank them for these insights. We’ve added to the introduction some information about retention in clinical care engagement and expanded in the discussion and conclusion how these strategies could be used in clinical care settings to support the clinical care continuum. 

In introduction: see lines: 110-113. 

In the discussion: 

Furthermore, given our high equitable retention using both well-documented and novel retention strategies, we hope this can be applied to clinical care retention. Prior studies have shown that Individuals with CL involvement and other competing needs to achieving good health (e.g. unstable housing) led to poor retention in clinical care—contributing to health inequities. 

In the conclusion: We hope these retention strategies can also be used to improve clinical care retention among this population. Furthermore, future research that documents the cost effectiveness of retention efforts is urgently needed. 

Tables 1 and 2 

Thorough but quite large, may be beneficial to fit each one onto one page 

We thank the reviewer for this recommendation. We’ve streamlined the language and have gotten Table 1 to one page and Table 2 to two pages. 

Methods 

Primary outcome is retention at 18 months, did virtual /phone visits count for this outcome? 

Yes virtual visits did count and we’ve added the following language (in red) to explain that: Study retention was defined as the proportion of study participants who completed the scheduled study visits (in-person or virtual) within the allowable follow-up visit window. 

All participants were on community supervision, was there any interaction with the study team and probation/parole officers? 

There was limited interaction with probation/parole officers during recruitment, for logistical scheduling purposes only. Probation/parole officers were never used for retention purposes because we wanted to ensure study participants’ participation or non-participation was not influenced by the criminal legal system. 

Results 

It would be nice to know which retention methods were deployed and how often. Presumably the staff retention training and methods occurred for all staff, but did all participants receive all the retention strategies? As these are labor-intensive endeavors, it would be helpful to know what was truly required to achieve this level of retention. This could help guide future efforts, whether they be in research or for the purposes of providing healthcare or other post-incarceration services, in order to estimate what level of resources are needed to provide these services. Also, some individuals may need less, some may need more engagement to be retained and it would be useful to understand this spectrum. This is reminiscent of a “bundled” approach to quality improvement where it is difficult to tease out what was most effective. Did everyone start out with the same level of retention strategies deployed and then adapted based on need or was is consistent throughout regardless of engagement? The penultimate paragraph of the discussion (is this a limitations paragraph?) addresses this to some degree but would be helpful to describe in methods and/or results how these strategies were deployed to the extent that this was measured. 

We thank the reviewer for these thoughtful comments. And we agree completely that this is a limitation of the current study. We have added a “Limitations” section and added the following sentences (in red below) to explain this limitation and the future opportunity to be able to measure retention efforts using our new app-based system. 

Limitations 

While we do believe quality and quantity of retention strategies contributed to retaining 92.3% of our sample, it can be challenging to quantify these efforts, particularly by each retention strategy. Prior work has repeatedly concluded that it is not possible to identify which specific retention strategies were particularly helpful in retention when multiple were used (25). While calls to action for more evaluative retention data have been made, this has largely not been done. Our retention efforts for this cohort wave (e.g., those that enrolled prior to the COVID-19 pandemic) were conducted primarily via paper and pencil or locked, fillable PDFs. While all retention strategies used for each participant were documented in the retention journal, the analysis of which retention methods were most successful is too labor intensive to quantify. This is a limitation of this study, as we are unable to analyze the “dose” of retention activities and the amount of time required to retain the most difficult cases. To that end, we have recently developed an app-based recruitment and retention tracking system for our second cohort wave (e.g., those that enrolled in the study after the COVID-19 pandemic began). This platform allows for detailed documentation of contact attempts (i.e., which phone numbers were previously used) and the amount of time spent on each activity by research staff. The platform will allow us to run real-time frequencies of, for example, the number of changes to participant locator information throughout the study and the number of attempts by retention strategy type, (e.g., mailings, phone calls to participant contact) that lead to successful retention. This system will increase data fidelity and contribute to the research gap to quantify retention efforts. Further, the retention field would benefit from qualitative insights from participants as to which strategies most contributed to their study and/or clinical retention. 

Discussion 

Would benefit from having an actual limitations paragraph. If the authors are not able to measure which retention strategies were deployed when or some type of “dose” of retention intervention, then this needs to be explicitly stated as a limitation. Also, it may have been helpful to hear from participants what they felt contributed most to retention (either through a survey or qualitative interviews), as this too could help guide future efforts. 

Yes, we agree with the reviewer and have added a limitations paragraph to address these two points. See above paragraph 

Reviewer 2: 

Reviewer #2: This is an important contribution to a growing body of work that highlights the value of including people with criminal-legal (CL) involvement in research, and strategies to keep participants engaged in the midst of a global pandemic. This is particularly important for populations with overlapping disparities, such as those with CL involvement. I strongly recommend publication, however I have a few suggestions for revisions below: 

1) Introduction (Line 72): You discuss retention strategies for criminal-legal (CL) involved individuals, however I wonder if you can include some reasons for loss to follow-up as well (for example, historical mistrust of researchers). Additionally, a brief discussion of trends in retention for different sociodemographic/geographic groups (or lack thereof) would be helpful as this is a component of your analysis. 

We thank the reviewer for their comment and have added the following: 

Historically, research was designed with little to no input from racial and ethnic minorities and other groups experiencing health inequities. Mistrust around research participation is rooted in structural racism and justifies hesitancy to participating and staying engaged in research studies. In a meta-analysis of 165 studies focused on recruitment and/or retention techniques of low-income or minority populations, Nicholson and colleagues (2015) found only 15 (9%) studies focused on retention (10). Of these, language barriers and mistrust were citated as common barriers to retention. 

2) Line 147: What informed your decision to use these specific retention strategies? Including the citation from Table 1 would be helpful here that provides some context and reasoning. 

Thank you for this great idea. We added some text to explain that retention strategies were identified via an in-depth literature search and the experience and expertise of the authors from prior retention studies. We also made more clear in the narrative that the five categories were informed by the citation in Table 1. 

3) Line 151: Generally, across what time period were changes to your retention strategies implemented during the COVID-19 pandemic? As written, the reader is left to infer it was around the time of March 2020. Including a timeline provides more support for your claim that these strategies contributed to a high retention rate. 

We have added the dates to the table as to when each retention activity was approved and introduced. 

4) Tables 1 and 2: Different verb tenses and writing styles are used throughout the table in an inconsistent manner. I recommend streamlining them to reflect an objective description of the strategies throughout. I also recommend removing any strategies that were not used in this study from the table instead of denoting them as "Strategy not used in this research study". Strategies that were not used can be disclosed in the manuscript text, however I don't think it is necessary. 

We thank the reviewer for this comment. We’ve gone through and made sure verb tenses agree and reduced the text to only reflect strategies that were used in this study. We did not disclose them in the manuscript text as we agree with the reviewer and do not think that is needed.

---

## [Editor Report · Decision Letter 1]

13 Mar 2023

Retention Strategies among those on Community Supervision in the South: Lessons Learned During the COVID-19 Pandemic

PONE-D-22-30143R1

Dear Dr. Uhrig Castonguay,

We’re pleased to inform you that your manuscript has been judged scientifically suitable for publication and will be formally accepted for publication once it meets all outstanding technical requirements.

Kind regards,

Douglas S. Krakower, MD

Academic Editor

PLOS ONE
---

## [Editor Report · Acceptance letter]

22 Mar 2023

PONE-D-22-30143R1 

Retention Strategies among those on Community Supervision in the South: Lessons Learned During the COVID-19 Pandemic 

Dear Dr. Uhrig Castonguay:

I'm pleased to inform you that your manuscript has been deemed suitable for publication in PLOS ONE. Congratulations! Your manuscript is now with our production department. 

Kind regards, 

on behalf of

Dr. Douglas S. Krakower 

Academic Editor

PLOS ONE